# Overcoming Resistance to Checkpoint Inhibitors with Combination Strategies in the Treatment of Non-Small Cell Lung Cancer

**DOI:** 10.3390/cancers16162919

**Published:** 2024-08-22

**Authors:** Amanda Reyes, Ramya Muddasani, Erminia Massarelli

**Affiliations:** Department of Medical Oncology & Therapeutics Research, City of Hope National Medical Center, Duarte, CA 91010, USA; amareyes@coh.org (A.R.); rmuddasani@coh.org (R.M.)

**Keywords:** immune checkpoint inhibitors, antibody–drug conjugates, bispecific antibodies, dual checkpoint blockade, tumor-infiltrating lymphocytes, neoantigen vaccines, immunotherapy resistance, combination therapy, PD-L1 inhibitors

## Abstract

**Simple Summary:**

Treatment options for lung cancer have greatly expanded in the last decade. Immune checkpoint inhibitors for patients without driver mutations have become the standard of care in widespread disease and, more recently, localized disease. However, resistance can develop, and new treatments are needed. We discussed current combination treatments with radiation and chemotherapy, as well as newer agents.

**Abstract:**

Lung cancer continues to contribute to the highest percentage of cancer-related deaths worldwide. Advancements in the treatment of non-small cell lung cancer like immune checkpoint inhibitors have dramatically improved survival and long-term disease response, even in curative and perioperative settings. Unfortunately, resistance develops either as an initial response to treatment or more commonly as a progression after the initial response. Several modalities have been utilized to combat this. This review will focus on the various combination treatments with immune checkpoint inhibitors including the addition of chemotherapy, various immunotherapies, radiation, antibody–drug conjugates, bispecific antibodies, neoantigen vaccines, and tumor-infiltrating lymphocytes. We discuss the status of these agents when used in combination with immune checkpoint inhibitors with an emphasis on lung cancer. The early toxicity signals, tolerability, and feasibility of implementation are also reviewed. We conclude with a discussion of the next steps in treatment.

## 1. Introduction

According to the WHO, lung cancer is the leading cause of cancer-related mortality worldwide [1]. Non-small cell lung cancer (NSCLC) contributes to the vast majority of lung cancer diagnoses 85% vs. 15% small cell lung cancer. NSCLC can be further categorized by histology into non-squamous (largely adenocarcinoma) 78% and squamous 18% with a low percentage of rare histologies such as adenosquamous [2]. Recently, the rates of squamous carcinoma have been decreasing in developed countries, reflecting higher rates of smoking cessation [3]. In the last two decades, the treatment landscape for NSCLC has changed after the discovery of immune checkpoint inhibitors (ICIs) and targetable ‘driver mutations’ [4]. Furthermore, agents that modulate the immune system (immunotherapy) include vaccines (both dendritic cells and peptides), cytokines, and oncolytic viruses, among others, with varying success in lung cancer [5].

ICIs are a type of immunotherapy directed at promoting the immune system’s innate ability to attack cancer cells. ICIs target specific proteins known to inhibit the immune response such as Program Cell Death Protein 1 (PD-1), Programmed Cell Death Ligand 1 (PD-L1), and Cytotoxic T-Lymphocyte Associated Protein 4 (CTLA-4). By blocking the proteins, they restore the immune system’s ability to target cancer cells [6,7]. ICIs, specifically pembrolizumab, atezolizumab, and cemiplimab, were proven to be effective first-line agents in Programmed Cell Death Ligand 1 (PD-L1)-positive metastatic NSCLC in phase-3 trials [8,9,10]. Combination PD-L1 and cytotoxic T-lymphocyte-associated protein 4 (CTLA-4) inhibitors, nivolumab and ipilimumab, were also proven to be effective in the first-line compared to standard of care chemotherapy, regardless of PD-L1 status [11]. Our review aims to focus on therapies currently approved or under study in combination with ICIs.

Despite the widespread success and proven efficacy of checkpoint inhibitors in lung cancer, limitations have been observed. Driver mutations, namely EGFR and ALK, have been largely excluded from clinical trials with checkpoint inhibitors [8,9,10,11,12]. Of note, oncogenic-mutation-driven NSCLC (driver mutations) comprises approximately 60–70% of cases. The vast majority are Kirsten Rat Sarcoma viral oncogene homolog [KRAS], followed by Epidermal Growth Factor Receptors (EGFRs) [13]. Several mechanisms have been proposed as to the lack of efficacy of ICIs in general in oncogene-driven NSCLC, including a low tumor mutational burden and low immunogenicity [14,15]. Even among lung cancer patients without driver mutations, only 40% of patients demonstrated a clinically significant response [16]. Of the initial responders to ICIs, both primary and secondary resistance remains a concern [17].

Primary resistance is defined by the Society for Immunotherapy of Cancer (SITC) as patients who have disease progression within less than 6 months after receiving at least 6 weeks of immune checkpoint inhibitors [18]. There are several proposed mechanisms for primary and secondary resistance including both intrinsic and extrinsic factors. Mutations in the IFN-y pathway, increased neoantigen intratumor heterogeneity, low mutational burden, transcriptomic features, epigenetic modifications, mutations in Beta 2-microglobulin and deficiencies in HLA antigen presentation have been previously described as intrinsic to resistance to ICIs [19,20,21,22,23,24,25] (Figure 1). Meanwhile, extrinsic factors such as low tumor infiltrating lymphocytes (immune cold tumors), the upregulation of alternative immune checkpoint receptors, and tumor microenvironment factors have also been observed [26,27,28]. Even patient-specific factors like concurrent medications and gut microbiome play a role in ICI resistance [29,30]. This review aims to investigate the mechanisms of resistance and potential strategies to overcome this resistance with combination therapies including radiation, chemotherapy, dual immunotherapy, vaccines, and antibody–drug conjugates, among others.

## 2. Checkpoint Inhibitors and Chemotherapy

In the current treatment of lung cancer, the combination of ICIs and chemotherapy has been successful. The combination was established in the first-line setting in metastatic NSCLC across PD-L1 positivity [31,32,33,34,35] (Table 1). There were some differences across trial designs, with IMPOWER130, KEYNOTE-198, and EMPOWER-Lung 3 evaluated single immune checkpoint blockade with chemotherapy, while POSEIDON and CHECKMATE 9LA evaluated dual blockade in combination with chemotherapy [31,32,33,34,35]. Interestingly, when evaluating specific response by histology (adenocarcinoma, squamous, adenosquamous, etc.), only cemiplimab and nivolumab/ipilimumab showed improved PFS and OS inpatients with squamous histology from diagnostic biopsy alone compared to the intention to treat population, which included several pathologic subtypes [33,35].

After evaluation in the metastatic setting, investigators sought to explore the use of various immune checkpoint inhibitors in the perioperative setting in resectable NSCLC. These studies focused on resectable disease, including stages IB–IIIB, but the vast majority included stages II–IIIA [36,37,38,39,40,41,42] (Table 1). Patients with N3 disease were excluded from the perioperative clinical trials and only a portion of the trials included stage-IIIB patients at a lower percentage of the study population with 14.5% in KEYNOTE 671, 21% in NADIM II, and in 25% in AEGAN [36,37,38,39,40,41,42] (Table 1).

Surprisingly, when pembrolizumab was utilized in the adjuvant setting for up to 18 cycles in stage-IB–IIIA disease, less benefit was seen in the PD-L1 > 50% cohort than the general patient population [42] (Table 1). The addition of pembrolizumab to chemotherapy in the perioperative setting improved survival and treatment response, such as the major pathological response rate of 30.2% compared to 11.0% in the placebo group (95% CI, 13.9 to 24.7; *p* < 0.0001) and pathological complete response of 18.1% compared to 4.0% (95% CI, 10.1 to 18.7, *p* < 0.0001) [36]. Nivolumab in combination with chemotherapy was first evaluated in the neoadjuvant setting in stage-IB–IIIA resectable disease, with improved median event-free survival of 31.6 months (95% CI 30.2 to not reached) compared to 20.8 months (95% CI, 14.0 to 26.7) (HR 0.63; 95% CI, 0.43 to 0.91; *p* = 0.005) with chemotherapy alone [37] (Table 1). Later, nivolumab was tested in the perioperative setting in stage-IIIA/B disease with neoadjuvant nivolumab and chemotherapy followed by 6 months of adjuvant nivolumab in patients with R0 resections, with improvement in both complete pathologic response rates and PFS [38]. Similar efficacy was seen with other agents in this class, such as durvalumab and atezolizumab [39,40,41].

## 3. Combination Immunotherapy

Dual immune checkpoint inhibitor blockade has been proposed as a potential avenue to combat both lower response rates and resistance to immunotherapy. Notably, in patients with negative PDL1 status, defined as <1%, the combination of PD-L1 and CTLA-4 inhibitors (nivolumab plus ipilimumab) had an improved median duration of response of 17.2 months (95% CI, 12.8 to 22.0) compared to 12.2 months (95% CI, 9.2 to 14.3) with standard of care chemotherapy [11]. A recent meta-analysis found that compared to chemotherapy, dual immunotherapy had improved outcomes in both overall survival (HR = 0.76, 95% CI: 0.69–0.82) and progression-free survival (HR = 0.75, 95% CI: 0.67–0.83) across all levels of PD-L1 expression, including negative (PDL1 < 1%) to high (PDL1 > 50%) [43]. However, caution should be used prior to the utilization of this method as some combination ICI treatments, namely pembrolizumab/ipilimumab and durvalumab/tremelimumab, had increased toxicity and did not improve PFS and OS [44,45]. The difference in response among combinations of dual checkpoint inhibitors could be due to a variety of factors including differences in the inhibitors themselves and trial design with placebo vs. chemotherapy control arms, among others [44,45]. Although dual checkpoint inhibitor therapy improves nonresponse, it does not completely solve the problem, which suggests other mechanisms of resistance via which cancer cells evade the immune system.

Atezolizumab was further investigated in combination with chemotherapy and bevacizumab in the IMPOWER150 study, which had an increased median OS of 19.2 months with atezolizumab vs. 14.7 months (HR 0.78 95% CI 0.64 to 0.96; *p* = 0.02) with bevacizumab and chemotherapy [46]. Newer agents, known as novel checkpoint inhibitors, are currently under exploration. One such agent includes T-cell receptor with immunoglobulin and immunoreceptor tyrosine-based inhibitory motif domain [TIGIT] monoclonal antibodies. Tiragolumab (Anti-TIGIT antibody) in combination with ICIs (atezolizumab) demonstrated a clinical benefit in PD-L1 > 1% NSCLC with an ORR of 31.3% (95% CI 19.5–43.2) compared to 16.2% (95% CI 6.7–25.7, *p* = 0.031) in the single-agent atezolizumab group and a slight benefit in progression-free survival of 5.4 (95% CI 4.2-not estimable) vs. 3.6 months (2.7–4.4 months) in the single-agent atezolizumab group [47] (Table 2). SKYSCRAPER-01 seeks to further understand the role of combination IO and TIGIT targeting in a randomized phase-3 study evaluating tiragolumab plus atezolizumab vs. atezolizumab [48].

Lymphocyte-Activation Gene 3 (LAG-3), another agent, is a protein that is believed to downregulate T helper 1 cell activation, proliferation, and cytokine secretion. High levels of LAG-3 expression correlate with tumor progression and a poorer prognosis. LAG-3 along with PD-1 dual blockade have been shown in mouse models to improve anti-tumor immune response by increasing CD8+ tumor-infiltrating cells in the tumor microenvironment and decreasing T regulatory cells [49]. Eftilagimod alpha (LAG-3 protein) in combination with pembrolizumab in a phase-II study (TACTI-002) has been shown to achieve a response in a PD-L1 unselected metastatic non-small cell lung cancer population in the first-line setting. In this single-arm study, response rates for squamous and non-squamous pathology were 33.3% and 40.3%, respectively [50].

Yet another novel immune modulator under investigation is utomilumab, a human IgG2 monoclonal antibody agonist of the T-cell costimulatory receptor 4-1BB, which was evaluated in a phase-1b study that demonstrated utomilumab in combination with pembrolizumab was well tolerated and demonstrated anti-tumor activity in advanced solid tumors, including NSCLC. In this phase-1b study, twenty-three patients received a combination of utomilumab with pembrolizumab, of which six patients were found with a confirmed complete or partial response [51] (NCT02179918). OX40, also known as CD134 or tumor necrosis factor receptor superfamily membrane, is a type I transmembrane glycoprotein expressed by T-cells. OX40 has been detected on tumor infiltrating lymphocytes in NSCLC and its ligand (OX40L) on cancer cells [52,53,54]. In a phase-I clinical trial, the combination of GSK3174998, a novel humanized IgG1 monoclonal antibody agonistic specific for OX40, was investigated in patients with advanced solid tumors with or without pembrolizumab, but the results showed limited clinical activity [55] (NCT02528357).

Combination immune checkpoint inhibitors with vascular endothelial growth factor (VEGF) receptor inhibition have been known to demonstrate responses in multiple solid tumor types including renal cell carcinoma, endometrial carcinoma, and hepatocellular carcinoma. In a phase-II trial, Lung-MAP substudy S1800A, the combination of ramucirumab with pembrolizumab vs. SOC was tested in advanced NSCLC patients that previously received PD-1 or PD-L1 therapy and progressed after at least 84 days. The study demonstrated improved overall survival of 14.5 months (80% CI 13.9–16.1) vs. 11.6 months (80% CI 0.9–13.0) in the standard-of-care group [56]. Based off this, S2302 Pragmatica-Lung has been developed, which randomizes previously treated stage-IV or recurrent NSCLC patients who have progressed on IO after at least 84 days to SOC vs. ramicirumab plus pembrolizumab; enrollment is ongoing [57]. As demonstrated, combination immunotherapy has emerged as a promising approach in treating lung cancer. By targeting multiple ICIs such as PD-1, PD-L1, and CTLA-4 simultaneously, this strategy aims to enhance the immune response against cancer cells and potentially unleash a stronger and more durable response. Ongoing research and clinical trials continue to explore the optimal combinations and sequences of immunotherapeutic agents to maximize efficacy while minimizing side effects.

## 4. Checkpoint Inhibitors and Radiation

Radiation has been an essential treatment modality in lung cancer for decades, especially for localized control. However, the first observation of the distant effects of radiation (outside the radiation field) was first noted in the 1950s and described as the ‘abscopal effect’ [58]. This was further expanded on in the setting of ICIs with the discovery that radiated tumors cells release tumor-associated antigens that can further activate CD8+ T-cells by a variety of mechanisms including increasing the antigen-presenting dendritic cell presentation of T-cell receptors [TCR], thereby augmenting the adaptive immune response of CD8+ T-cells [59] (Figure 2). Radiation also serves to upregulate the expression of MHC (major histocompatibility complex) class I, which is often downregulated by tumors as a mechanism to evade immune detection [60,61] (Figure 2). Interestingly, pre-clinical studies found a lack of the abscopal effect in immune-deficit environments in comparison to immunocompetent environments [62].

This concept was translated into clinical practice in metastatic NSCLC patients in the PEMBRO-RT and MD Anderson Cancer Center (MDACC) trials evaluating treatment with pembrolizumab with or without radiotherapy [63,64]. One key difference between the trials was PEMBRO-RT only included previously treated patients, while MDACC included patients who had not received previous chemotherapy. A pooled analysis of these trials found the best out-of-field abscopal response rate (ARR) was 19.7% with pembrolizumab compared to 41.7% (OR 2.96, 95% CI 1.42 to 6.20; *p* = 0·0039) with pembrolizumab plus radiotherapy, and the median overall survival was 8·7 months (95% CI 6.4 to 11.0) with pembrolizumab alone compared to 19.2 months (95% CI 14.6 to 23.8) (HR 0.67 0.54 to 0.84; *p* = 0·0004) with pembrolizumab plus radiotherapy [65].

Combination durvalumab and radiation was also evaluated in the locally advanced setting in *Phase II Study of Durvalumab (MEDI4736) Plus Concurrent Radiation Therapy in Advanced Localized NSCLC Patients* (DOLPHIN trial), with a median PFS of 25.6 months (95% CI, 13.1 months to not estimable) at a median follow-up of 22.8 months and ORR of 90.9% (95% CI, 75.7 to 98.1) [66]. Of note, grade-3 or -4 pneumonitis was observed in 11.8% of patients receiving combination therapy. Additionally, the addition of durvalumab also improves outcomes in non-resectable disease when added to definitive chemoradiation, with an improvement in OS of 47.5 months with durvalumab compared to 29.1 months with placebo and an estimated 5-year overall survival rate of 42.9% with durvalumab versus 33.4% [67].

Oligometastatic progression in NSCLC remains a concern that has prompted further investigation. Radiation, particularly high-dose radiation, with stereotactic ablative radiotherapy (SBRT) has been used to treat oligometastatic disease, with improvements in outcomes including PFS and OS on long-term follow-up [68,69]. Additionally, the use of ICIs, namely pembrolizumab for a minimum of eight cycles, after the completion of radiation treatment for oligometastatic disease has improved survival; on long-term follow-up, the median PFS was 19.7 months (95% CI 7.6 to 31.7) compared to a historical median of 6.6 months (determined by [70]) (*p* = 0.005) and the OS at 5 years was 60.0% (SE, 7.4%) [71,72].

In addition to augmenting the ICI response by increasing the presentation of tumor-associated antigens, radiation may play a larger role in resistance. Several studies have demonstrated the various roles of radiation in mitigating ICI resistance, notably by altering the tumor microenvironment [73,74,75]. Along with tumor microenvironment, pre-clinical studies found higher levels of oxidative phosphorylation (OXPHOS) contributes to ICI efficacy and as radiation can increase OXPHOS levels which may be a mechanism of improving ICI sensitivity [76,77]. Furthermore, there appears to be a synergistic effect with the combination of radiation and anti-CTLA4, whereby anti-CTLA4 inhibits Tregs and radiation increases the TCR exposure to diverse tumor antigen [59,78]. This may help explain why the addition of radiation improves the checkpoint inhibitor response in PD-L1-low or tumor mutational burden (TMB)-low disease [79]. Additional investigations into combination radiation and ICI may further elucidate the role of radiation in resistance.

## 5. Checkpoint Inhibitors and Antibody–Drug Conjugates

Antibody–drug conjugates and bispecific antibodies have entered the forefront of treatment across malignancies and lung cancer is no exception. Early studies found microtubule inhibitors similar to the agents utilized in brentuximab vedotin induce the maturation of dendritic cells and increase tumor-presenting antigens, thereby priming T-cells and enhancing the response to immune-directed therapies [80]. Similar immune-modulating affects were also seen with pyrrolobenzodiazepine dimer or tubulysin payloads of antibody–drug conjugates [81]. Enapotamab vedotin, an antibody–drug conjugate that targets AXL receptor tyrosine kinase, has been shown to overcome resistance to ICIs/immune-mediating cell killing and stimulate an inflammatory response, thereby inducing cytotoxic T-cells in both melanoma and lung cancer models [82]. Additionally, pre-clinical studies of a bispecific antibody targeting both transforming growth factor-beta (TGFB) and PD-L1 showed anti-tumor activity in vitro and in vivo [83].

Various clinical trials are evaluating antibody–drug conjugate/bispecific antibody combinations with ICIs in solid tumors and in lung cancer specifically. The MORPHEUS Lung trial is evaluating the combination of atezolizumab and sacituzumab govitecan, while another trial is utilizing pembrolizumab and trastuzumab deruxtecan in metastatic NSCLC (NCT03337698 and NCT04042701, respectively) [84] (Figure 3). Other ongoing trials are investigating triplet combinations of ADCs, ICIs, and chemotherapy in various settings, including the upfront setting, such as ADVANZAR (durvalumab) with datopotamab deruxtecan, TROPION-Lung02 (pembrolizumab) with datopotamab deruxtecan, and EVOKE-02 (pembrolizumab) with sacituzumab govitecan (NCT05687266, NCT04526691, and NCT05186974, respectively) [85,86] (Figure 3).

Increasingly adverse events related to these combination therapies remain a concern in the KATE02 trial in breast cancer, with 33% adverse events compared to 19% in the control arm [87]. However, a review of safety profiles of early-phase studies with other ADCs, including trastuzumab deruxtecan, did not find higher rates of interstitial lung disease (ILD) with the addition of ICIs [88]. The results of these upcoming trials will provide valuable insights on the efficacy and safety of these combination treatments.

## 6. Checkpoint Inhibitors and TILs

Adoptive cell therapies (ACTs) consist of three subtypes of treatment known as tumor-infiltrating lymphocytes (TILs), T-cell-receptor-engineered T-cells (TCT-Ts) and chimeric antigen receptor T-cells (CAR-Ts) [89,90,91]. To date, these treatments remain in clinical trials for the treatment of NSCLC. Of the current ACTs conducted in NSCLC, there are studies underway to evaluate the combination of TIL therapy with checkpoint inhibitors. TIL therapy was initially discovered in 1988 by Dr. Steven Rosenberg and involves the expansion of T-cells found in harvested tumor tissue followed by reinfusion [92]. TILs have been studied in various solid tumors, showing the most promise in melanoma, with lifiluecel having been approved recently. There are many challenges when considering TIL therapy, including but not limited to isolating tumor tissue, expanding the T-cell population in vitro, and improving the time needed for production, on average 6–8 weeks. Additionally, there are multiple factors attributed to resistance with TILs. Some known resistance mechanisms include internal factors, specifically TIL quiescent genes, and external factors, including autologous tumor cells that inhibit TILs [93]. Studies are underway to determine if combining ICIs with TILs to alter the PD1/PD-L1 axis may overcome these resistance mechanisms.

Most recently, in the NSCLC arena, clinical trials are underway to assess the combination of ICIs with TIL therapy. Nivolumab in combination with TIL was conducted as a phase-1 single-arm study by Creelan et al. [94]. Patients received four cycles of nivolumab before TIL infusion. Following lymphodepletion chemotherapy, TILs and IL-2 infusions were given to 16 patients with proven progression. Thereafter, patients received nivolumab every 4 weeks for up to a year. Of the 20 patients on trial, 16 received TIL therapy, of which 13 were evaluated for response. Radiographic responses occurred in 6 of the 13 evaluable patients, of which 2 patients had complete responses that remained ongoing 1.5 years later (NCT03215810). Another trial looked at the combination of autologous TIL (LN-145) in combination with durvalumab; however, this was withdrawn due to increasing adverse toxicities from IL-2 administration (NCT03419559) [95]. The STARLING trial is evaluating the combination of TBio-4101 (TIL) and pembrolizumab, while another trial currently recruiting is combining L-TIL plus tislelizumab for PD1 antibody-resistant NSCLC (NCT05576077 and NCT05878028, respectively) [96,97]. The results of these clinical trials and further investigation into the combination of ICIs with TIL therapy will provide a better understanding of the safety and efficacy profile.

Despite the success of TILs in melanoma, we remains unclear on its efficacy in NSCLC [98]. Notable disadvantages of TILs include the timeframe required to start treatment. This includes the successful resection and expansion of TILs, which can take up to 8 weeks, during which patients can develop a heavy disease burden. Patients also require an adequate performance status with good cardiac and pulmonary reserves, which may be difficult to achieve in patients with a heavy disease burden [90].

## 7. Checkpoint Inhibitors and Vaccines

Genetic alterations in the DNA of cancer cells result in the production of altered peptide sequences called tumor-specific antigens (TSAs), also known as neoantigens [99,100,101]. These neoantigens are not expressed on normal tissues. As such, neoantigen-directed vaccines are an exciting therapeutic focus as they provide a tumor-directed immune response [102,103]. The tumor mutational burden (TMB) and the neoantigen load are, in general, associated with an overall increase in the anti-tumor activity of checkpoint inhibitors. Clinical trials aim to combine neoantigen-directed vaccines with ICIs based on the rationale that this will increase T-cell activity against these neoantigens [104]. Given NSCLC is known to have a higher TMB compared to other solid tumors, the focus on combining ICIs with neoantigen-directed vaccines may prove beneficial.

Here, we discuss clinical trials underway to further understand the combination of neoantigen-directed vaccines with ICIs. Neoantigens are tumor-specific antigens created from somatic gene mutations solely expressed in tumor tissues. These trials are currently in the recruitment stage and data are yet to be reported. NCT03715985 is a clinical trial evaluating EVAX-01-CAF09b, a neoantigen vaccine containing up to 15 peptides derived from somatic mutations from a patient’s individual cancer with CAF09b, a liposomal adjuvant delivery system, in combination with ICIs [105]. NEO-PV-01 is a neoantigen vaccine of up to 20 peptides, which is currently being study in administration with pembrolizumab and chemotherapy [104] (NCT03380871). Also being conducted is a trial of a personalized and adaptive neoantigen vaccine (PANDA-VAC), which consists of up to nine tumor-specific antigens, in combination with pembrolizumab (NCT04266730) in squamous cell lung cancer [106]. Additionally, there is the PNeoVCA study including a personalized neoantigen peptide-based vaccine of up to 20 peptides focused on both class-I and class-II human leukocyte-antigen-bound neoantigens, which are thought to be more potent neoantigen candidates in combination with pembrolizumab (NCT05269381) [107].

In addition to the neoantigen vaccines noted, there are other types of vaccines that have been under study. Clinical trial NCT02879760 is a phase-I study focusing on understanding the efficacy of oncolytic MG1-MAGEA3 with Ad-MAGEA3 vaccine in combination with pembrolizumab in NSCLC. MG1MA3 is a Maraba virus modified to express tumor antigen MAGE-A3. It is felt that MG1MA3 after immune priming with MAGE-A3-modified adenovirus may trigger anti-tumor T-cell responses [108].

The data from these trials are much anticipated as these vaccines are noted to have a tumor-directed focus, thereby reducing toxicity as they do not produce an autoimmune response [109]. With advances in molecular sequencing technology with both whole exome and RNA sequencing, the prediction of tumor-specific antigens for the creation of neoantigen-directed vaccines has improved, thus making the process of developing these therapies more feasible [110]. This personalized therapeutic approach will hopefully yield increased efficacy, especially in combination with ICIs.

## 8. Discussion

While ICIs have revolutionized the treatment of lung cancer in the last decade, limitations remain even among the patients who respond initially. A method to combat both primary and secondary resistance is the addition of combination agents, including dual ICIs, chemotherapy, radiation, and, more recently, antibody–drug conjugates/bispecific antibodies, TILs, and vaccines. The majority of these modalities have proven efficacy in various lines of treatment in combination with ICIs. The adoptive cellular therapies, antibody–drug conjugates, and neoantigen vaccines represent a more personalized therapeutic approach in the age of precision medicine. However, additional research is required to further stratify the sequencing of these treatments and determine the role of each specific treatment modality in various lines of treatment, including the perioperative setting. In addition, there should be increased attention on adverse events and potential additive toxicities such as pneumonitis, which was seen in previous sequencing of EGFR tyrosine kinase inhibitors and ICIs [111]. In addition to known adverse events, these novel therapeutics like ADCs can create unique toxicities, including dermatologic, ophthalmologic, and neurologic toxicities, requiring specialty care, which may be limited in certain practice settings [88]. Central nervous system toxicity or immune-cell-associated neurotoxicity (ICANS) seen with cellular therapies is a concern with the addition of TILs and certain bispecific antibodies and may limit the utilization of these therapies in patients with pre-existing CNS disease.

From a more pragmatic approach, the costs of these novel treatments, many of which require hospital admission for monitoring during the first cycle at minimum, should be considered as this could be prohibitive in certain practice settings. To better inform treatment selection and determine which patients would derive the greatest benefit from the utilization of these novel agents, new biomarkers would be beneficial as, from our current data, PD-L1 appears to be an imperfect marker. Serine–threonine kinase liver kinase B1 (STK11), first discovered as the cause of Peutz–Jeghers syndrome, upregulates the AMP-Kinase pathway, thereby contributing to resistance [112,113,114]. Kelch-like ECH-associated protein 1 (KEAP1) forms part of E3 ubiquitin ligase, which regulates transcription factor NF-E2-related factor 2 (NRF2), and has an important role in responding to oxidative stress, and has been to linked to degenerative disease as well as an immunosuppressive tumor microenvironment with low infiltrating CD8+ T-cells [115,116]. Both STK11 and KEAP1 predict worse a prognosis and treatment response to ICIs, most notably in combination with KRAS mutations [114,117,118].

The landmark trials on ICIs excluded patients with EGFR mutations, but from other smaller studies and smaller subgroup analysis, ICIs have been shown to be ineffective in these patients, regardless of PDL1 status [119,120,121,122]. However, it should be mentioned that not all driver mutations seem to behave the same as EGFR, ALK, and HER2, as BRAF, MET, and KRAS may have some benefit from ICIs [7,123]. There are several theories for this observation including changes in the tumor microenvironment, low immunogenicity, and low infiltrating lymphocytes [119]. As per our research, there is one trial to date within the KRAS-oncogene-driven population studying the combination of checkpoint inhibitors with sotorasib in KRAS-inhibitor-naïve patients with a KRAS G12C mutation. The results showed a higher incidence of grade 3–4 immune-related adverse events, most notably, elevated liver enzymes [124]. Understanding the role of combination checkpoint inhibitor therapy with targeted therapy in selected driver mutated populations will require further study. The greatest benefit could potentially be seen with MET-targeted antibody–drug conjugates and ICIs [123].

## 9. Conclusions

ICIs will remain a cornerstone of treatment in NSCLC, in both early stage and metastatic disease. We remain hopeful that combination therapy with ICIs will prove fruitful in overcoming resistance mechanisms and provide patients with a lasting treatment response. In the future, combination treatment with ICIs and another novel agent may even lead to the heavily sought after outcome of curing metastatic disease. The results and completed data analysis of the studies discussed will provide the necessary information regarding these novel therapeutic approaches in clinical practice and guide future treatment paradigms.

## Figures and Tables

**Figure 1 cancers-16-02919-f001:**
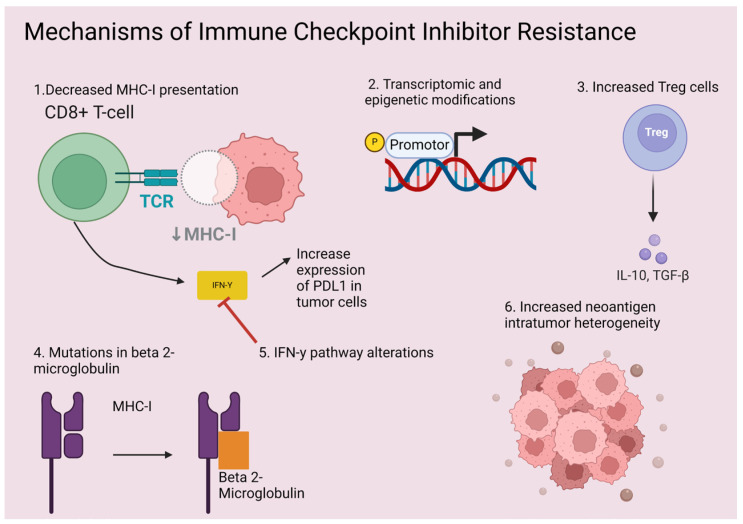
Mechanisms of immune checkpoint inhibitor resistance. Created with BioRender.

**Figure 2 cancers-16-02919-f002:**
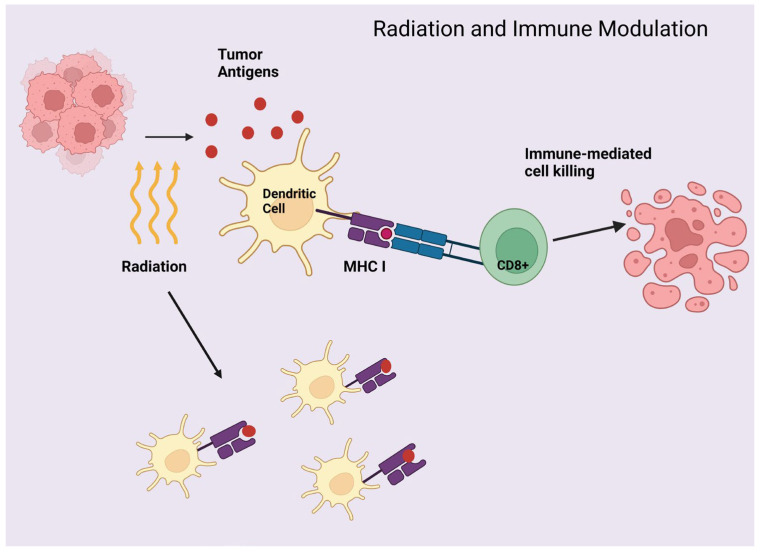
Radiation and immune modulation. Created with BioRender.

**Figure 3 cancers-16-02919-f003:**
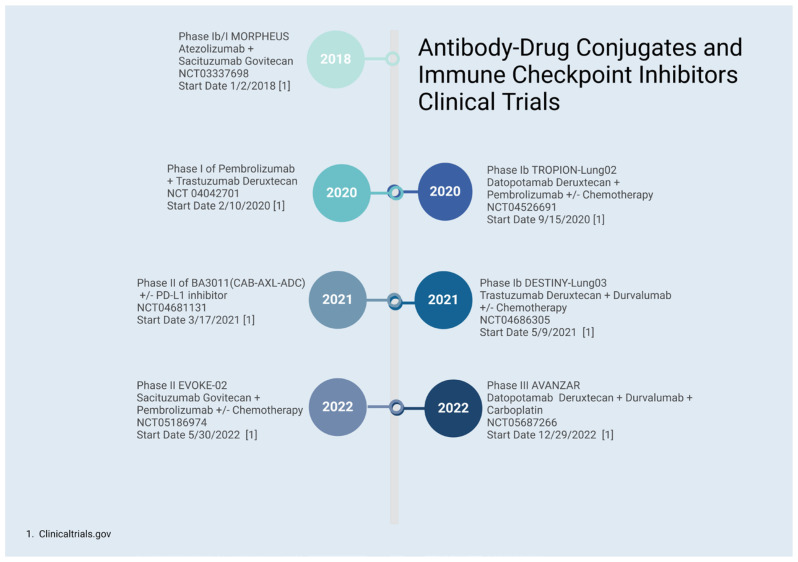
Immune checkpoint inhibitors and antibody–drug conjugates. Created with BioRender.

**Table 1 cancers-16-02919-t001:** Immune checkpoint inhibitors + chemotherapy in NSCLC.

Immune Checkpoint Inhibitor	FDA Approved Metastatic PD-L1 > 1% without Chemo ^1^	FDA Approved Metastatic PD-L1 > 50% without Chemo ^1^	FDA Approved Metastatic Any PD-L1 + Chemo ^1^	Neoadjuvant	Adjuvant	Duration of Adjuvant	Survival Analysis	MDR and % Grade 3–4 Adverse Events (AE)
Pembrolizumab	Yes	Yes	Yes	Yes, Stage II–IIIB KEYNOTE-671	Yes, Stage IB–IIIA KEYNOTE-091	Up to 54 weeks KEYNOTE-091 Up to 39 weeks KEYNOTE-671	KEYNOTE 198: PFS 8.8 months with Pembro + chemo vs. 4.9 in chemo + placebo; OS not reached in Pembro + chemo vs. 11.3 months in chemo + placebo 091: DFS 73.8% vs. 63.1% in placebo 671: EFS 62.4% pembro + chemo vs. 40.6% in chemo only OS 80.9% vs. 77.6% in chemo only	198: AEs 67.2% in Pembro + chemo vs. 65.8% chemo + placebo 091: AEs 34.3% vs. 25.7% in placebo 671: AES 44.9% vs. 37.3% in chemo only
Nivolumab	No	No	No	Yes, Stage IB–IIIA CheckMate-816	Yes, Stage IIIA/B NADIM II	6 months	816: DFS 76.1% Nivo + chemo vs. 63.4% chemo only NADIM: PFS 67.2% Nivo + chemo vs. 40.9% chemo only OS 85.0% in Nivo + chemo vs. 63.6% in chemo only	816: AEs 33.5% vs. 36.5% in chemo alone NADIM: AEs 22% in Nivo + chemo vs. 10% in chemo only
Nivolumab and Ipilimumab	Yes	Yes	Yes	Stage I–IIIA NCT03158129	N/A	N/A	CheckMate 227: PDL1 > 1%: PFS 17.1 months Ipi/Nivo vs. 14.9 months with chemotherapy; 2-year OS 40.0% with Ipi/Nivo vs. 32.8% chemo only 9LA: PDL1 < 1%: OS Ipi/nivo +chemo 16.8 months vs. 9.8 months chemo only	227: MDR 17.1 months vs. 13.9 months in chemo only; AEs 32.8% in Ipi/Nivo vs. 36.0% in chemo 9LA: AEs: 47% in Ipi/nivo + chemo vs. 38% in chemo alone
Atezolizumab	No	Yes	Yes	Yes, Stage IB–IIIB IMpower 030	Yes, Stage II–IIIA IMpower 010	12 months	IMPOWER 010: DFS 56% in atezo vs. 49% in placebo 110: OS PDL > 50% 20.2 months in atezo vs. 13.1 months chemo 132: PFS 7.6 months Atezo chemo vs. 5.2 months with chemo; OS 17.5 months Atezo chemo vs. 13.6 months chemo	010: AEs 22% in atezo vs. 12% in placebo 110: AEs: 30.1% with atezo vs. 52.5% in chemo 132: AEs 54.6% Atezo + chemo and 40.1% chemo
Durvalumab	No	No	No, PDL1 > 1%	Yes, Stage II–IIIB AEGEAN	Yes, Stage II–IIIB AEGEAN	12 months	AEGEAN: EFS 73.4% durvalumab vs. 64.5% placebo	AEGEAN: AEs 42.4% durvalumab vs. 43.2% with placebo
Cemiplimab	No	Yes	Yes	No	No	N/A	EMPOWER Lung 3: OS 21.9 months with cemiplimab plus chemo vs. 13.0 months chemo only 1: PDL1 > 50% PFS 8.2 months with cemiplimab vs. 5.7 months with chemo; OS not reached with cemi vs. 14.2 months with chemo	3: AEs: 43.6% with cemi + chemo vs. 31.4% chemo only 1: AEs 28% with cemiplimab vs. 39% with chemo

^1^ https://www.fda.gov/drugs/ accessed on 16 June 2024. N/A: Not applicable.

**Table 2 cancers-16-02919-t002:** Combination immunotherapy with novel immune checkpoint targets.

Line of Therapy	Agent	Immune Checkpoint Inhibitor	Clinical Benefit	Trial Name
Metastatic, first-line NSCLC	riragolumab (TIGIT)	atezolizumab	ORR of 31.3% (PD-L1 > 1%)	CITYSCAPE
Locally advanced unresectable stage-III NSCLC, maintenance	domvanalimab (TIGIT)	durvalumab	Ongoing	PACIFIC-8
Metastatic, first-line NSCLC	eftilagimod alpha (LAG-3)	pembrolizumab	RR 33.3% (squamous NSCLC) RR 40.3% (non-squamous NSLCL) PD-L1 unselected	TACTI-002
Metastatic, first-line NSCLC	utomilumab (4-1BB mAb agonist)	pembrolizumab	N/A	NCT02179918
Advanced solid tumors	GSK3174998 (OX40 agonist mAb)	pembrolizumab	Limited activity DCR 9% at ≥24 weeks	NCT02528357
Metastatic or recurrent NSCLC, PD after PD1 or PD-L1 inhibitor after ≥84 days	ramucirumab (VEGF inhibitor)	pembrolizumab	OS Benefit: 14.5 months (80% CI 13.9–16.1) vs. 11.6 months (80% CI 0.9–13.0)	Lung-MAP substudy S1800A
Metastatic or recurrent NSCLC, PD after PD1 or PD-L1 inhibitor after ≥84 days	ramucirumab (VEGF inhibitor)	pembrolizumab	Enrollment in progress	Pragmatica S2302
Metastatic non squamous NSCLC who had not received prior chemotherapy	bevacizumab	atezolizumab	OS Benefit: 19.2 months vs. 14.7 months (HR 0.78 95% CI 0.64 to 0.96; *p* = 0.02)	IMPOWER 150

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
