# Peer review of "Overcoming Resistance to Checkpoint Inhibitors with Combination Strategies in the Treatment of Non-Small Cell Lung Cancer"

_cancers, 2024, doi:10.3390/cancers16162919_

Round 1
Reviewer 1 Report
Comments and Suggestions for Authors
General comments:
The aim of this study was to review various therapy combination strategies, immunotherapy combination strategies, in the treatment of NSCLC, type of lung cancer. However, the manuscript is very hard to follow and has many deficiencies because of which I have to recommend rejection!
Major comments:
1. There is no definition/explanation of any essential term to follow this manuscript that the authors classify as a REVIEW......The review always provides short educational explanations of elementary concepts with which the authors subsequently deals. Without proper introduction and short explanations, good definitions this paper (REVIEW) has neither foundation nor supporting walls. For example, what is immunotherapy, why we use it, what kinds of immunotherapy exist...???? Therefore, the reader is put in a position to guess the meanings, and then explanations.
2. Reading this REVIEW, one can get the impression that IMMUNOTHERAPY is exclusively and only blockade of immune checkpoints (with agents of the complicated names that are properly but not fully listed). However, the authors nowhere, absolutely nowhere, not even in passing, mention that monoclonal antibodies (At) are behind those names that bind to membrane proteins (receptors) whose role in a normal situation is to stop the Immune response/reaction, which is why we call them control points (immune reaction brakes). Also, the authors do not state clearly anywhere (only in one third of some sentence in the introduction shyly mention), that the blockade of those so-called "control points" (membrane proteins/receptors) practically reactivates the immune system. For a good review paper we would have to read about all immunotherapy treatment options and then we would know that:
3. TILs, Antibody Drug Conjugates, Cancer Vaccines... are different types of immunotherapy (Liu C, Yang M, Zhang D, Chen M and Zhu D (2022) Clinical cancer immunotherapy: Current progress and prospects. Front. Immunol. 13:961805.doi: 10.3389/fimmu.2022.961805 etc). Therefore, subtitles like "Immunotherapy and Antibody Drug Conjugates", "Immunotherapy and TILs", "immunotherapy and vaccines" can put the reader in a position to conclude that Antibody Drug Conjugates, TILs and Cancer Vaccines are additional non-immunological therapeutic approaches like Radiation or Chemotherapy. Wrong!!!
4. Most important, nothing about NSCLC!!!??? Except that it is a leading cause of cancer related deaths. Any information, clinical or molecular, about it!!!!????
Minor comments:
1. Trial names with numbers in the text are not compatible with those in the tables.
2. Language needs proofreading.
Comments on the Quality of English LanguageMinor lenguage editing.
Author Response
Reviewer 1:
The aim of this study was to review various therapy combination strategies, immunotherapy combination strategies, in the treatment of NSCLC, type of lung cancer. However, the manuscript is very hard to follow and has many deficiencies because of which I have to recommend rejection!
Major comments:
- There is no definition/explanation of any essential term to follow this manuscript that the authors classify as a REVIEW......The review always provides short educational explanations of elementary concepts with which the authors subsequently deals. Without proper introduction and short explanations, good definitions this paper (REVIEW) has neither foundation nor supporting walls. For example, what is immunotherapy, why we use it, what kinds of immunotherapy exist...???? Therefore, the reader is put in a position to guess the meanings, and then explanations.
We agree with the reviewer and have provided a more detailed explanation of immunotherapy in the background. Additionally, to provide further clarity immunotherapy is now clearly defined throughout the paper including the subtitles.
- Reading this REVIEW, one can get the impression that IMMUNOTHERAPY is exclusively and only blockade of immune checkpoints (with agents of the complicated names that are properly but not fully listed). However, the authors nowhere, absolutely nowhere, not even in passing, mention that monoclonal antibodies (At) are behind those names that bind to membrane proteins (receptors) whose role in a normal situation is to stop the Immune response/reaction, which is why we call them control points (immune reaction brakes). Also, the authors do not state clearly anywhere (only in one third of some sentence in the introduction shyly mention), that the blockade of those so-called "control points" (membrane proteins/receptors) practically reactivates the immune system. For a good review paper we would have to read about all immunotherapy treatment options and then we would know that:
We agree with the reviewer that additional background information is important for understanding the concept. We included explanation and relevant definitions as indicated.
- TILs, Antibody Drug Conjugates, Cancer Vaccines... are different types of immunotherapy (Liu C, Yang M, Zhang D, Chen M and Zhu D (2022) Clinical cancer immunotherapy: Current progress and prospects. Front. Immunol. 13:961805.doi: 10.3389/fimmu.2022.961805 etc). Therefore, subtitles like "Immunotherapy and Antibody Drug Conjugates", "Immunotherapy and TILs", "immunotherapy and vaccines" can put the reader in a position to conclude that Antibody Drug Conjugates, TILs and Cancer Vaccines are additional non-immunological therapeutic approaches like Radiation or Chemotherapy. Wrong!!!
We agree that the subsection titles can be confusing, and therefore we have changed the subsection titles to allow more clarity. Immunotherapy will now be defined clearly every time it is used.
- Most important, nothing about NSCLC!!!??? Except that it is a leading cause of cancer related deaths. Any information, clinical or molecular, about it!!!!????
We agree that an adequate review requires some information regarding lung cancer. We have added additional information to provide the necessary background.
Minor comments:
- Trial names with numbers in the text are not compatible with those in the tables.
We have cross checked the trial numbers in the table and text.
- Language needs proofreading.
We have reviewed the language and made corrections as necessary.

Reviewer 2 Report
Comments and Suggestions for Authors
This is a short but quite comprehensive review of combination strategies in overcoming resistance to current ICI treatment in advanced NSCLC.
To improve the readibility of the paper, a figure/cartoon summarizing the mechanisms of resistance to ICIs in NSCLC could be useful.
On the other hand, the authors should discuss in a dedicated paragraph the combination of ICIs with targeted agents in selected driver mutated populations (as mentioned in the paper discussion section).
Author Response
Reviewer 2:
This is a short but quite comprehensive review of combination strategies in overcoming resistance to current ICI treatment in advanced NSCLC.
- To improve the readability of the paper, a figure/cartoon summarizing the mechanisms of resistance to ICIs in NSCLC could be useful.
We agree with the review that a figure summarizing the mechanisms of resistance is helpful for reader understanding. We have created a figure depicting mechanisms of resistance and moved the discussion of resistance to the introduction.
- On the other hand, the authors should discuss in a dedicated paragraph the combination of ICIs with targeted agents in selected driver mutated populations (as mentioned in the paper discussion section).
We agree with the reviewer that this topic warrants further discussion. We have added an additional paragraph dedicated in the discussion of the combination of ICIs with targeted agents. This paragraph discusses both the current NSCLC clinical trial studying ICI in combination with sotorasib in the KRAS G12C mutated population and an explanation on the varying efficacy seen with ICIs in selected driver mutation populations.

Reviewer 3 Report
Comments and Suggestions for Authors
This is a well written review on the combinations of immunotherapy with other therapeutical strategies to overcome the (primary and secondary) resistance to immunotherapy of NSCLC.
The review is quite concise; has not unnecessary information and I read it with interest and have learned something from this lecture.
I have some remarks that may be considered to improve further this article:
1. This is about the treatment strategies to overcome the resistance to immunotherapy of patients treated for NSCLC, as stated in the aim of the study, discussion, and every chapter describing respective treatment combinations. Why not to indicated this in the title of the article? This would be even more interesting for a reader and will reflect a real content of the article. Current title is too general, in my opinion.
2. When describing the role of combination of radiation with immunotherapy, the indication for treatment with radiation of oligometastatic disease or oligoprogression or oligopersistence managed with immunotherapy should be at least mentioned. In these clinical scenarios, radiation may eradicate the gross foci of the disease, potentially eliminating resistant clones. It lacks of this relevant aspect of RT-ICI combination in the current review.
Author Response
Reviewer 3:
This is a well written review on the combinations of immunotherapy with other therapeutical strategies to overcome the (primary and secondary) resistance to immunotherapy of NSCLC.
The review is quite concise; has not unnecessary information and I read it with interest and have learned something from this lecture.
I have some remarks that may be considered to improve further this article
- This is about the treatment strategies to overcome the resistance to immunotherapy of patients treated for NSCLC, as stated in the aim of the study, discussion, and every chapter describing respective treatment combinations. Why not to indicated this in the title of the article? This would be even more interesting for a reader and will reflect a real content of the article. Current title is too general, in my opinion.
We agree with the reviewer and have changed the title to “Overcoming Resistance to Checkpoint Inhibitors with Combination Strategies in the Treatment of Non-Small Cell Lung Cancer.”
- When describing the role of combination of radiation with immunotherapy, the indication for treatment with radiation of oligometastatic disease or oligoprogression or oligopersistence managed with immunotherapy should be at least mentioned. In these clinical scenarios, radiation may eradicate the gross foci of the disease, potentially eliminating resistant clones. It lacks of this relevant aspect of RT-ICI combination in the current review.
We agree with the reviewer and have included a discussion about the treatment of oligometastatic disease in the manuscript.

Reviewer 4 Report
Comments and Suggestions for Authors
This review focuses on a very important topic of lung cancer immunotherapy, namely the combination of immunotherapy with other different types of treatment. The review discusses recent advances in the various combination treatments such as chemotherapy, immunotherapy, radiation, antibody-drug conjugates, bispecific antibodies, neoantigen vaccines, and TILs. The review also provides a discussion of recent advancements on the detection of early toxicity signals, tolerability, and feasibility of treatments. Potential future directions in combinatory therapy of lung cancer are also discussed.
Specific comments:
1. This review requires significant editorial work. F. e., “ Interestingly, in subgroup analysis only cemiplimab and nivolumab/ipilimumab showed continued benefit among squamous histology”. What type of subgroups? What kind of benefits? What is “squamous histology”? Be more specific and provide sufficient details. Next sentence: “This was followed by the more recent use in the perioperative setting in resectable NSCLC; the included stages varied by study but stage IIIB N2 disease was the highest allowed”. What exactly was followed? The use of cemiplimab or nivolumab/ipilimumab or both? Why did stages varied and how? In how many studies a Stage III B N2 was present? Sufficient details play an important role to understand the review content.
2. Table 1 is missing very important information on primary endpoints (overall survival, progression-free survival, etc.) and key findings (durability, adverse events, and benefits) after immunotherapy. Some terms in this Table have a superscript 1 without any further explanation. The use of bevacizumab is described in text but not shown in Table.
3. Provide full definitions for IMPOWER130, KEYNOTE-198, EMPOWER-Lung 3 abbreviations.
4. Combination immunotherapy. Specify the levels of PD-L1 expression considered as to be low or high, or any other grading used for evaluation of its levels of expression. Explain, why pembrolizumab/ipilimumab and durvalumab/tremelimumab combinations showed increased toxicity without survival benefits. Explain the alternative immune modulatory functions. Tiragolumab is probably not TIGIT. What the response rate for squamous and non-squamous pathology were compared to (shown percentage was compared to what?)? What anti-tumor activity utomilumab in combination with pembrolizumab demonstrated?
5. Table 2 is missing information on NCT02528357.
6. Immunotherapy and radiation. Irradiated tumor cells. What type of dendritic cell function? Provide full definitions for MDACC and DOLPHIN abbreviations. Table 3 should be shown based on the studies and results discussed in this section. Presentation of tumor-associated antigen by what type of cells to what type of cells with what type of results. How radiation alters tumor microenvironment? Why the addition of radiation may even improve immunotherapy response in PD-L1 low or tumor mutational burden (TMB) low disease? Some sections of this review do not have the appropriate visual representations. This section could benefit from having a figure illustrating the positive and negative effects of radiation on tumor cells and TME in terms of the use of a combination therapy with immunotherapeutic agents.
7. Figure 1 is of a low quality.
8. To what extent the efficacy of TILs in NSCLC remains questionable?
9. The authors show patient’s numbers in some discussed here clinical trials but omit them in others. Similar applies to the immunotherapy cycles.
10. Provide more details on NPC-sa001-CAF09b, L1 [NCT03715985], NEO-PV-01 with pembrolizumab and chemotherapy [NCT03380871]. What kind of neoantigen? What kind of vaccine?
11. Several protein-specifc vaccines used in NSCLC such as the CIMAvax epidermal growth factor (CIMAvax-EGF) vaccine, MAGE-A3, NY-ESO-1 and the BLP25 liposome vaccine (anti-MUC1) are missing here.
12. Discussion. Is the early and late resistance the same as the primary and secondary resistance? If not, explain the difference. Explain what are STK11 and KEAP1 and how they were established as biomarkers and for what type of cancer.
Author Response
Reviewer 4: .
This review focuses on a very important topic of lung cancer immunotherapy, namely the combination of immunotherapy with other different types of treatment. The review discusses recent advances in the various combination treatments such as chemotherapy, immunotherapy, radiation, antibody-drug conjugates, bispecific antibodies, neoantigen vaccines, and TILs. The review also provides a discussion of recent advancements on the detection of early toxicity signals, tolerability, and feasibility of treatments. Potential future directions in combinatory therapy of lung cancer are also discussed.
Specific comments:
- This review requires significant editorial work. F. e., “ Interestingly, in subgroup analysis only cemiplimab and nivolumab/ipilimumab showed continued benefit among squamous histology”. What type of subgroups? What kind of benefits? What is “squamous histology”? Be more specific and provide sufficient details. Next sentence: “This was followed by the more recent use in the perioperative setting in resectable NSCLC; the included stages varied by study but stage IIIB N2 disease was the highest allowed”. What exactly was followed? The use of cemiplimab or nivolumab/ipilimumab or both? Why did stages varied and how? In how many studies a Stage III B N2 was present? Sufficient details play an important role to understand the review content.
We agree with the reviewer that more detail in this section is required for adequate understanding. We included further details regarding the different pathology types. The second sentence needed more clarification as we were referring to the investigation of immune check point inhibitors in general. We also included a more detailed description of the advanced stages included in the various clinical trials and the percentages of Stage IIIB patients that were included.
- Table 1 is missing very important information on primary endpoints (overall survival, progression-free survival, etc.) and key findings (durability, adverse events, and benefits) after immunotherapy. Some terms in this Table have a superscript 1 without any further explanation. The use of bevacizumab is described in text but not shown in Table.
We agree with the review and have added this missing information to the table. We added information regarding Bevacizumab to Table 2 as we believed this was the more appropriate table for this trial.
- Provide full definitions for IMPOWER130, KEYNOTE-198, EMPOWER-Lung 3 abbreviations.
We appreciate the reviewer’s comment regarding the full definitions of the major trials. However, we believe the complete trial names would be more burdensome rather than adding clarification.
- Combination immunotherapy. Specify the levels of PD-L1 expression considered as to be low or high, or any other grading used for evaluation of its levels of expression. Explain, why pembrolizumab/ipilimumab and durvalumab/tremelimumab combinations showed increased toxicity without survival benefits. Explain the alternative immune modulatory functions. Tiragolumab is probably not TIGIT. What the response rate for squamous and non-squamous pathology were compared to (shown percentage was compared to what?)? What anti-tumor activity utomilumab in combination with pembrolizumab demonstrated?
We added the grading details in regard to the PD-L1 status. We added further possible explanation on why the combinations increased toxicity without survival benefits We deleted the term “Alternative immune modulatory functions” as this was unclear. We corrected the reference that Tigagolumab is not TIGIT but rather an anti-TIGIT antibody. We have clarified that TACTI-002 was a single arm study as such there are no response rates for comparison. We clarified that anti-tumor activity utomilumab in combination with pembrolizumab demonstrated a a complete or partial response in six of the twenty three patients included in this phase 1b study.
- Table 2 is missing information on NCT02528357.
We agree with the reviewer and have updated the table to include this trial.
- Immunotherapy and radiation. Irradiated tumor cells. What type of dendritic cell function? Provide full definitions for MDACC and DOLPHIN abbreviations. Table 3 should be shown based on the studies and results discussed in this section. Presentation of tumor-associated antigen by what type of cells to what type of cells with what type of results. How radiation alters tumor microenvironment? Why the addition of radiation may even improve immunotherapy response in PD-L1 low or tumor mutational burden (TMB) low disease? Some sections of this review do not have the appropriate visual representations. This section could benefit from having a figure illustrating the positive and negative effects of radiation on tumor cells and TME in terms of the use of a combination therapy with immunotherapeutic agents.
We thank the reviewer for these comments and have included further clarification regarding the type of dendritic function, the full definitions for the abbreviations and further discussion regarding antigen presentation and tumor microenvironment. We added discussion on how radiation improves response in PD-L1 low disease. We added a figure to demonstrate radiation effects on immune mediated cell killing.
- Figure 1 is of a low quality.
We agree with the reviewer and have redownloaded the image in higher quality.
- To what extent the efficacy of TILs in NSCLC remains questionable?
We have edited the sentence to explain the challenges behind TILs as a therapy noting the time and effort required for tissue procurement, expanding the T-cell population in vitro and the time needed for production (6-8 weeks). Additionally, we have discussed potential resistance mechanisms to TILs therapy.
- The authors show patient’s numbers in some discussed here clinical trials but omit them in others. Similar applies to the immunotherapy cycles.
We agree with the reviewer and have reviewed the results presented. For the trials for which data is available we have added the specific numbers. Please note as some of the clinical trials are in progress, the data is pending for these trials.
- Provide more details on NPC-sa001-CAF09b, L1 [NCT03715985], NEO-PV-01 with pembrolizumab and chemotherapy [NCT03380871]. What kind of neoantigen? What kind of vaccine?
We agree with the reviewer and have explained neo-antigen therapy in greater detail. Additionally, each neo-antigen vaccine under study and it’s unique properties are discussed.
- Several protein-specifc vaccines used in NSCLC such as the CIMAvax epidermal growth factor (CIMAvax-EGF) vaccine, MAGE-A3, NY-ESO-1 and the BLP25 liposome vaccine (anti-MUC1) are missing here.
We agree with the reviewer and have added the therapies that have been studied in combination with immunotherapy, which include MAGE-A3, CIMAvax epidermal growth factor. We have held off on including the BLP25 liposome vaccine as it does not appear to have been studied in combination with immunotherapy.
- Discussion. Is the early and late resistance the same as the primary and secondary resistance? If not, explain the difference. Explain what are STK11 and KEAP1 and how they were established as biomarkers and for what type of cancer.
We appreciate the reviewer’s comments regarding the different terminology for resistance, as the terms were intended to indicate the same concept, we have changed the terminology, so it is unified. We provided additional details regarding STK11 and KEAP1.

Round 2
Reviewer 4 Report
Comments and Suggestions for Authors
No further comments